# Quantitative optical assessment of photonic and electronic properties in halide perovskite

Adrien Bercegol[1,2], Daniel Ory[1,2], Daniel Suchet[2,3], Stefania Cacovich [2], Olivier Fournier[1,2], Jean Rousset[1,2] & Laurent Lombez [2,3]

The development of high efficiency solar cells relies on the management of electronic and optical properties that need to be accurately measured. As the conversion efficiencies increase, there is a concomitant electronic and photonic contribution that affects the overall performances. Here we show an optical method to quantify several transport properties of semiconducting materials and the use of multidimensional imaging techniques allows decoupling and quantifying the electronic and photonic contributions. Example of application is shown on halide perovskite thin film for which a large range of transport properties is given in the literature. We therefore optically measure pure carrier diffusion properties and evidence the contribution of optical effects such as the photon recycling as well as the photon propagation where emitted light is laterally transported without being reabsorbed. This latter effect has to be considered to avoid overestimated transport properties such as carrier mobility, diffusion length or diffusion coefficient.

[1] EDF R&D, 30 RD 128, 91120 Palaiseau, France. [2] IPVF, Institut Photovoltaïque d'Ile-de-France, 30 RD 128, 91120 Palaiseau, France. [3] CNRS, Ecole Polytechnique, Institut Photovoltaïque d'Ile-de-France UMR 9006, 30 RD 128, 91120 Palaiseau, France. Correspondence and requests for materials should be addressed to L.L. (email: laurent.lombez@cnrs.fr)

Photovoltaic (PV) devices offer a direct conversion of the light source in electricity, providing a much-needed solution to meet climate targets and move towards a low-carbon economy. Over the last decades, we have witnessed a continuous increase of their power conversion efficiency (PCE), leading to record PCE values of 26.6% for crystalline silicon solar cells, of 28.8% for thin film technologies (GaAs) and of 23.3% for emerging PV such as perovskite solar cells[1,2]. However, to approach the theoretical limit, which is about 31% in classical devices[3], it is crucial to work on the electronic and optical performances, both affecting the output voltage and the current. In particular, a high voltage is directly linked to a high electron and hole electro-chemical potential splitting (i.e., the quasi-level Fermi splitting) across the whole cell thickness. In order to provide new insights for a better understanding of solar cells working principles, it is thus fundamental to measure the recombination mechanisms as well as the carrier diffusion length of the absorber material[4]. As most PV materials gain in quality, the photon recycling (PR i.e., self-absorption of the radiative emission) plays an important role as it can further boost the carriers' concentration, so the voltage[5]. Besides, high current requires good carrier transport properties such as high carrier mobilities.

Among the photovoltaic technologies, perovskite solar cells have shown an unprecedented rapid increase of the conversion efficiency[6,7]. The remarkable photovoltaic performances might originate from a long carrier diffusion length and an efficient photon recycling[8,9] mechanism becoming an important contribution to the voltage enhancement. Depending on the measurement techniques as well as the film fabrication routes, there are differences in reported charge carrier diffusion lengths with values varying from the micrometer to the millimeter range[10–12]. These huge variations in term of transport properties can be ascribed to the quality of perovskite absorbers, to their chemical composition and to the grade of crystallinity of the thin films[13]. However, the characterization approach plays also a key role in the determination of such material properties. Looking at steady-state photoluminescence (PL) images allows to assess the lateral diffusion length of charge carriers[14–16] and underline the transport anisotropy in unpassivated polycrystalline CH₃NH₃PbI₃ (MAPI) films[17]. Time-resolved diffusion profiles have also been observed following a local pulsed illumination[18–22] and the widening of the PL profiles give insight into the carrier mobility. Regarding the PR, several studies have underlined this optical process in perovskite absorber, either by focusing on the slow radiative recombination process[23–25], or by highlighting the long-range charge carrier regeneration[8,26,27]. However, a model able to decorrelate electronic and photonic contributions to transport is still missing.

Here we present a quantitative optical experiment, to measure unbiased optoelectronic properties and to separate the electronic and the photonic contribution. Our innovative experimental approach consists in imaging the carrier diffusion induced by a point illumination and in analyzing the time and spectral dependence of the luminescence signal. The use of complementary multidimensional imaging techniques allowed us to collect and analyze datasets of the luminescence signal, characterized by high spatial, spectral and temporal resolution. Additionally, we rigorously solve the complete time-dependent lateral diffusion and we highlight the presence of three contributions which we quantify: the photon recycling part, the photon propagation of weakly absorbed light and the pure electronic diffusion. The method can be applied to several semiconductor materials. In this work, we present a study on last generation triple cation perovskite thin film where we quantitatively assess the photon recycling and the radiative recombination, as well as the transport properties such as the carrier diffusion length and the carrier mobility.

## Results

**Multi-dimensional imaging techniques.** In this study, we have investigated a triple cation mixed halide perovskite thin film ($(MA_{0.14}FA_{0.86})_{0.95}Cs_{0.05}Pb(I_{0.84},Br_{0.16})_3$) spin coated on glass (see Fig. 1a). Further details on the material fabrication process and technical details on the experimental setups are provided in the methods section. Experiments are done at 300 K under ambient conditions in air atmosphere.

A pulsed laser ($\lambda = 550$ nm, $\Delta t = 6$ ps, $f = 1$ MHz) was focused on the sample, creating a local charge carrier generation. The carriers diffuse and eventually recombine radiatively yielding to a luminescence signal. The lateral diffusion was imaged by employing

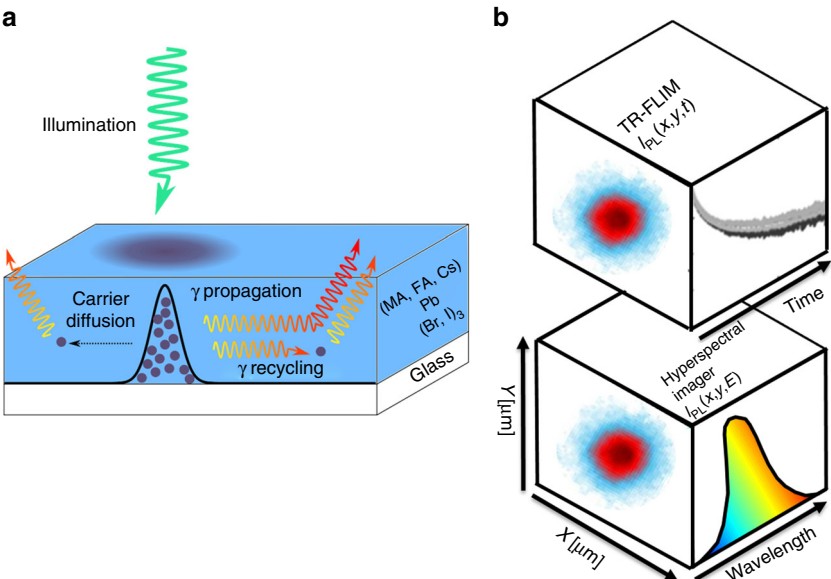

**Fig. 1** Sketch of the experimental observations and of the physical mechanisms taking place. **a** Sample structure and sketch showing the triple mechanism (carrier diffusion, photon propagation, and photon recycling) studied here **b** sketch showing the acquisition in 3 dimensions with time-resolved fluorescence imaging (TR-FLIM) or hyperspectral imaging (HI), for a point illumination sketched in the left face

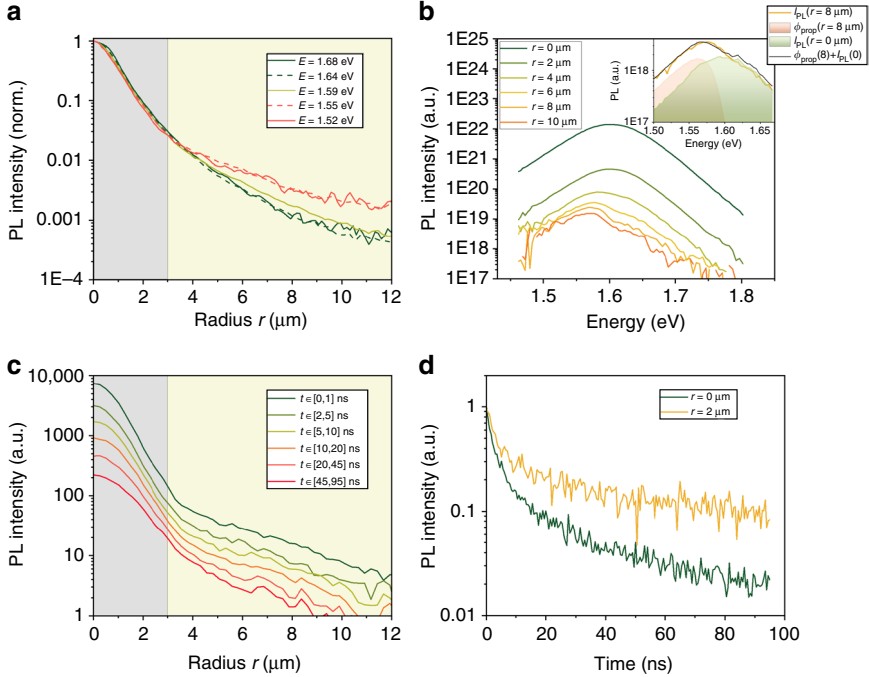

**Fig. 2** Experimental results obtained with the multidimensional imaging methods. **a** PL profiles for each emitted photon energy from 1.52 eV to 1.68 eV. The PL regime, where electronic diffusion dominate, appears in blue. The photon propagation regime, where photonic regime dominate, appears in green. **b** PL spectra at increasing distances from the laser spot. The inset shows $I_{PL}$ spectrum at $r = 8$ μm as the sum of a locally emitted and a propagated contribution. **c** PL profiles at incremental time steps after the laser excitation. **d** Time-resolved PL decay at the center, and at $r = 2$ μm, both normalized at $t = 0$ ns

two different experimental setups. First, an hyperspectral imaging system (HI)[28] (Fig. 1b) was used to record spectrally-resolved images with a spectral resolution of 2 nm. Then, through a time-resolved fluorescence imaging system (TR-FLIM)[29], we monitored the temporal evolution of the images with a time resolution of 750 ps. As the diffusion is isotropic from the local excitation spot, we averaged the recorded luminescence signal $I_{PL}$ in radial coordinates $r$ in order to significantly enhance our sensitivity. Two fluences were considered ($\phi_0 = 5 \times 10^6$ and $10^5$ photons per pulse).

**Evidencing the electronic and photonic regime**. We first look at the normalized spectrally-resolved PL profiles $I_{PL}(r)$ by using the HI. In Fig. 2a, they are drawn at different wavelengths as a function of the distance to the excitation point. We notice the presence of two different regimes. At a shorter distance, the PL spectra is independent from the photon energies $E_{photon}$. At a longer distance, it appears that $I_{PL}$ for the less energetic photons $E_{photon} < E_{peak}$ is maintained over a longer distance as compared to $E_{photon} > E_{peak}$; $E_{peak}$ being the emission energy at which the maximum PL intensity is reached. Indeed, the ratio between $I_{PL}(1.52\,\text{eV})$ and $I_{PL}(1.65\,\text{eV})$ increases (i.e., the red wavelengths of the PL signal propagate farther). Statistical analysis shows that 15% of the emitted photons diffuse on a distance larger than 10 μm in the perovskite. These long-range photons end up either by getting partially reabsorbed to excite charge carrier, which ultimately recombine (PR), or by being coupled out of the thin film. If all of them were reabsorbed, we would observe a local PL emission and expect no PL spectra variation (see Supplementary Fig. 11). However, the PL spectra at different distances from the excitation spot are presented in Fig. 2b and a spectral change for $r > 3$ μm is apparent. It indicates a strong contribution of a simple photon propagation at longer distance without contribution to the PR. The following expression of $I_{PL}$ summarizes the previous considerations and splits the monitored signal $I_{PL}$ between a propagated flux $\phi_{prop}$ which progressively red-shifts as $r$ increases,

and a local PL emission with a constant spectrum:

$$I_{PL} = \phi_{prop} + R_{eh}^* n^2 \qquad (1)$$

This interpretation was first introduced by Pazos-Outon et al.[8], and we confirm it by analytical calculations accounting for total internal reflexions of the isotropically emitted PL photons. The contribution of propagated PL photons is described analytically by $\phi_{prop}$ derived in Supplementary Note 3. We investigated the shape of PL spectra at each $r > 3$ μm, where their variation begins. We fit them as the linear combination of $I_{PL}$ spectrum observed at $r = 0$, representative for the local emission, and of a propagated spectrum at distance $r$. Results are displayed in the inset of Fig. 2b for $r = 8$ μm, and for multiple values of $r$ in Supplementary Fig. 5. The reproduction of monitored luminescence spectra is excellent. The relative weight of the direct emission is 100% at 3 μm and decreases to 40% at $r = 9$ μm, whereas the weight of propagated spectrum follows an opposite trend. This assesses the photon propagation regime and confirms the trapping of photoluminescence signal inside the thin film leading to photonic propagation and recycling over longer distance than what could be inferred from experiments realized on single crystals[26,30]

Table 1 summarizes previous considerations for the interpretation of spectrally-resolved measurements. In a nutshell, we can relate the $I_{PL}(r < 3\,\mu\text{m})$ to the local charge carrier concentration, while further spectral analysis is required to investigate transport properties using $I_{PL}(r > 3\,\mu\text{m})$. This spectral analysis indicates that $I_{PL}(E > E_{peak})$ can be ascribed to local PL emission, as $\phi_{prop}(E > E_{peak}) \approx 0$ for large values of $r$. This long-range charge carrier concentration is assumed to represent photon recycling, which will be justified with the next derived submicrometric electronic diffusion length. Hence, a characteristic length $L_{PR}$ for the photon recycling can be extracted by fitting the decay of $I_{PL}(r)$ for $E > E_{peak}$ with an exponential attenuation length. It decays with characteristic length of 2.5 μm. Knowing that the PL

**Table 1 Summary of the main physical contributions in the luminescence spatial variation**

| r: Distance to excitation spot | Spectrally-resolved $I_{PL}$, $I_{PL}(E<E_{peak})$ | Spectrally-resolved $I_{PL}$, $I_{PL}(E>E_{peak})$ | Time-resolved $I_{PL}$, $I_{PL}(t)$ |
|---|---|---|---|
| $r<3$ μm | $I_{PL} = R_{eh}^* n^2$ | $I_{PL} = R_{eh}^* n^2$ | $I_{PL} = R_{eh}^* n^2$ |
| $r>3$ μm | $I_{PL} = \phi_{prop} + R_{eh}^* n^2$ | $I_{PL} = R_{eh}^* n^2$ | $I_{PL} = \phi_{prop} + R_{eh}^* n^2$ |

Physical interpretation of the monitored signal $I_{PL}$ ($r$) using HI (column 2 & 3) or TR-FLIM (column 4). Two distinct zones appear, limitated by $r = 3$ μm. The photon propagation regime corresponds to $\phi_{prop}$ and local PL emission to $R_{eh}^* n^2$. The 3 μm value is defined empirically from the change in the PL profile in Fig. 2 and matches with the value of $1/\alpha$ at $E = 1.57$ eV

decreases quadratically with the carrier concentration $n$, $L_{PR}$ is measured around 5 μm. This length applies for photon recycling after a point pulsed illumination. This photonic transport is essentially lateral and probably affected by the film thickness.

We then explored the time dependence of the PL profile by using the TR-FLIM setup. Figure 2c shows the spectrally integrated time-resolved profiles while Fig. 2d shows the time-resolved luminescence at the central location $r = 0$ μm and away at $r = 2$ μm. We observe that the luminescence intensity $I_{PL}$ decreases faster at the center. At all times, the attenuation of $I_{PL}$ is slower at $r > 3$ μm than at $r < 3$ μm. This leads to a broadening of the PL profile as time goes on and constitutes a first hint towards electronic diffusion. Thanks to the aforementioned experimental results obtained with the HI, we infer that $I_{PL}$ profiles in Fig. 2c at $r > 3$ μm are hardly exploitable due to the concomitant contribution of PL emission and photon propagation $\phi_{prop}$. Hence, we will focus on the short distances $r < 3$ μm around the excitation spot to analyze the electronic contribution without being affected by the photonic propagation ($\phi_{prop} \approx 0$). The next section is dedicated to the description of the model able to quantify both the transport and the recombination phenomena. Specifically, precise and injection-independent values for the diffusion coefficient $D_n$, the internal radiative recombination coefficient $R_{eh}$, along with its correction factor to account for PR, the Shockley-Read-Hall lifetime $\tau_n$ will be calculated by solving time-dependent diffusion equations.

**Modeling results**. We model perovskite thin films as intrinsic semiconductors, where we can write as in Table 1:

$$\forall (\text{for all})t,\ r<3\mu m\ :\ I_{PL}(r,t) \propto n^2(r,t) \quad (2)$$

Non-radiative recombinations are considered monomolecular, and are described thanks to a Shockley-Read-Hall lifetime $\tau_n$, while radiative recombinations are modeled via the internal radiative coefficient $R_{eh}$[31]. Auger recombination are neglected as they take a significant share in the recombination process only for $n > 10^{18}$ cm$^{-3}$[32,33]. The local excitation creates a charge carrier gradient, which homogenizes through charge carrier diffusion[34] (diffusion coefficient $D_n$) as well as through photon diffusion[32]. The local reabsorption of this photon flux leads to the emergence of an additional generation term $g_{rec}$ in the continuity equation[23]. In the following, we will neglect the in-depth dimension to focus on the lateral transport. For this purpose, we will restrain our dataset to the long times, when in-depth diffusion has already happened (typically $t > 10$ ns for 600-nm-thick perovskite layers[35]). Therefore, the diffusion equation writes in polar coordinates:

$$\frac{\partial n}{\partial t}(r,t) = D_n \nabla n(r,t) - \frac{n(r,t)}{\tau_n} - R_{eh}n^2(r,t) + g_{rec}(r,t) \quad (3)$$

Under steady-state homogeneous illumination, the recycling is proportional to the PL emission and a single correction factor $g_{corr}$ can be applied on the measured external radiative coefficient value $R_{eh}^*$, to obtain the internal radiative coefficient $R_{eh}$[23,25,36]. It is defined as $g_{corr} = R_{eh}/R_{eh}^*$. However, this assumption does not hold in a more general case[32], in particular for transient

experiments or for inhomogeneous illumination conditions. As a consequence, we investigated the behavior of $g_{rec}$ with a radiometric model, depending on the charge carrier concentration profile. Electronic transport happens at the nanosecond scale and immediate photon propagation is hence assumed. Therefore, we write $g_{rec}$ as proportional to the PL emission around the considered point. This emission gets attenuated with an absorption coefficient $\alpha_{PR}$ (Beer-Lambert), and by a $1/r^2$ factor due its isotropic emission. It shall be noted that we considered here an average absorption coefficient $\alpha_{PR}$ defined in Supplementary Equation 7. This assumption is frequently used[27,36] and we show in the Supplementary Fig. 3 that it constitutes a valid approach at short distances from the excitation. When applied at long distances from the excitation (typically $r > 3$ μm), another expression for $g_{rec}$ accounting for variations of $\alpha(E)$ is used. Physical and geometrical details on the model, as well as the results are presented in the Supplementary Note 1–3, while we recall here the main output. It is summarized in Eq. (4), which allows to calculate $g_{rec}(r)$ by defining a secondary frame $(r',\theta',z)$ centered in $(r,0)$.

$$g_{rec}(r) = \iiint_{(r',\theta',z')\ \text{frame centered in}\ (r,0,0)} \frac{R_{eh}n^2(r',\theta')\alpha_{PR}}{4\pi} \frac{\exp(-\alpha_{PR}r')}{r'} dr'd\theta'dz' \quad (4)$$

Notably, it can be applied to any location of the thin film. One has to redefine the center of the polar frame at a point distinct from the center of the charge carrier distribution. We calculated $g_{rec}$ for Gaussian charge carrier distribution with full width at half maxima (FWHM) comprised between 1 and 3 μm, which are representative for the evolution of PL profiles between 10 and 100 ns. Outputs are displayed in Supplementary Fig. 2. For FWHM > 1.5 μm, the correction factor remains constant at $g_{corr} = 0.43 \pm 0.05$ over the first microns of the distribution. Thus, by spatially restraining our dataset to the first microns, it is possible to use a constant value for the external radiative coefficient $R_{eh}^*$ and to remove $g_{rec}$ from the continuity equation.

All previous considerations have led us to restrain spatially and temporally the dataset. Due to the photon propagation regime, we focus on $r < 3$ μm. Due to in-depth diffusion and inhomogeneous PR, we focus on $t > 10$ ns. Now, we can fit our time-resolved data, in order to derive unbiased values for the free parameters $D_n$, $\tau_n$ and $R_{eh}^*$. Figure 3a, b display two views from the fitted $I_{PL}(r,t)$ surface with Equation (3), along with experimental observations. Figure 3d displays the logarithmic difference between experimental and numerical values, which remains negligible at any fitting point. The good correlation between experimental and simulated results provides a clear evidence of the data consistency. Figure 3e–g characterize the sensitivity of the model to the fitting parameters $R_{eh}$, $D_n$, and $\tau_n$, by displaying the $I_{PL}(r,t)$ surface for various values around the fitted values. The variation of each parameter acts differently on the $I_{PL}(r,t)$ variation ensuring the independence of the determined parameters. The amplitude of the variations gives an estimation of the error for each parameter, also recalled in the next paragraph.

The diffusion coefficient $D_n$ is determined at $0.027 \pm 0.007$ cm$^2$ s$^{-1}$, the external radiative coefficient $R_{eh}^*$ is $4.4 \times$

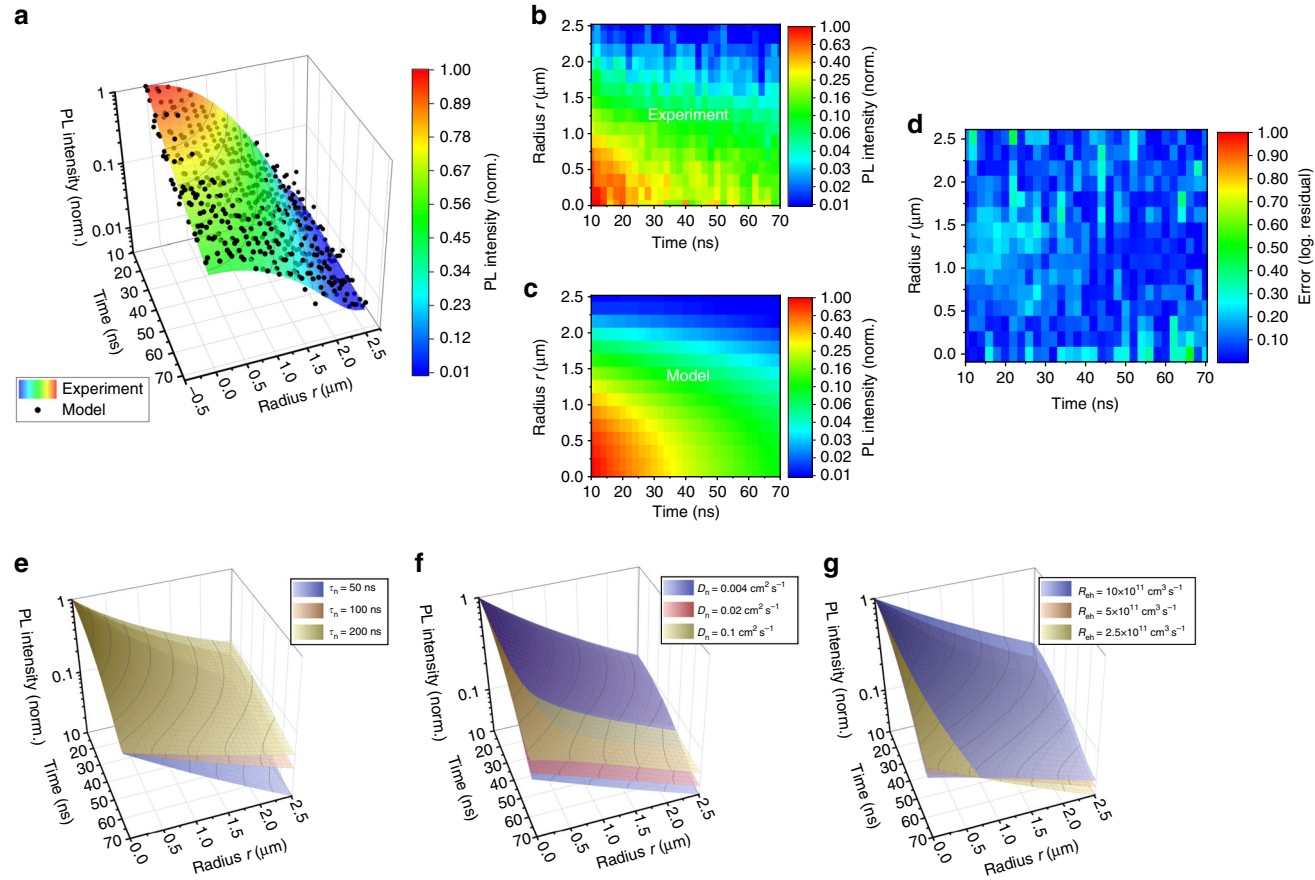

**Fig. 3** Determination of the pure electronic diffusion properties. **a** Surface showing the best-fit between $I_{PL}(r,t)$ calculated with the diffusion-recombination model in Equation (3) and experimental data (black circles) acquired on a mixed halide perovskite thin film ($\phi_0 = 5 \times 10^6$ photons per pulse). Fake colors indicate the PL intensity. **b** is a 2D map of the experimental data whereas **c** is a 2D map of the modeling results. **d** Logarithmic residuals map from the modeling. **e–g** Sensitivity analysis of the model to each parameter, displaying the $I_{PL}(r,t)$ for different values of $\tau_n$ (**e**) $D_n$ (**f**) and $R_{eh}^*$ (**g**) around the best fit parameters

$10^{-11} \pm 2 \times 10^{-11}$ cm³ s⁻¹, and the SRH lifetime $\tau_n$ is $110 \pm 30$ ns. Thanks to $g_{corr}$, we can also derive the internal radiative coefficient $R_{eh} = 1.0 \times 10^{-10} \pm 0.4 \times 10^{-10}$ cm³ s⁻¹. Auger recombination are not considered in our model and control fitting experiments to justify our approach are displayed in the Supplementary Fig. 9. An equivalent electronic diffusion length can be determined around 450 nm. It is here derived at $n = 10^{16}$ cm⁻³, which is relevant for PV operation at 1 sun excitation[37,38], and takes into account SRH and radiative recombinations for the definition of the lifetime. At such a fluence, the radiative lifetime is 660 ns, while the non-radiative one remains at 110 ns. The control experiment realized at low fluence (60 times less intense) is displayed in Supplementary Fig. 8. It gives similar result with $D_n$ at $0.035 \pm 0.02$ cm² s⁻¹ and $\tau_n$ at $90 \pm 30$ ns. The fit realized at low fluence is not sensitive to $R_{eh}^*$, which is consistent with the decreased weight of bimolecular recombination as $n$ decays. Control experiments also include fits realized without the time-domain restriction (Supplementary Fig. 6), or without the space domain (Supplementary Fig. 7). The first one is significantly less precise, while the latter leads to significant over-estimation of the diffusion coefficient, as photonic transport is also fitted with the electronic diffusion.

## Discussion
We have observed and analyzed diffusion profiles obtained for two injection levels and we have dissociated electronic diffusion, recycled and propagated photonic contributions. Electronic properties

are derived at short-range whereas photonic processes take place at a longer range. A full reconstruction of the time-resolved PL profiles allows quantifying the transport properties (diffusion coefficient, mobility, and diffusion length) as well as recombination-related parameters with no dependence on the injection level[39,40]. In our case, the different injection levels also underline the impact of PR. The quantitative assessment of this important factor allowed us to obtain a unique diffusion coefficient.

Modeling semiconductors transport properties often requires initial assumptions which need to be discussed. As slow diffusion happens in perovskite materials, in-depth diffusion could play a role at short times. In our case, we ruled out this effect, as the data analysis was performed on long time PL profiles. Second, trapping and detrapping events from shallow traps are neglected in our first order approach. These might induce a quicker PL quenching far from the excitation spot, where empty traps could remain. This could explain the low value derived for $\tau_n$, in comparison with previous studies accounting for traps[11,35,41]. Hopping mechanism from trap to trap might also contribute to the transport and explain the low values determined for $D_n$. Third, ambipolar effects are not considered, as electron and hole have similar mobilities in metal halide perovskite[42,43]. Finally, charge carrier extraction in photovoltaic devices based on perovskite layers works out of diffusion[44] and hence internal electric fields play a negligible role. We also recall that our measurement was applied on neat films, in which no built-in electric field due to heterointerfaces with transport layers is likely to happen.

This study also addressed the measurement of optical properties related to photon propagation inside a thin film. Previous models based on ray optics[23,24] or detailed balance considerations[25,45] had allowed to quantify the recycling process at the device scale for a homogeneous illumination. We employed inhomogeneous illumination to monitor transport and hence we needed a more precise model. The one we developed places itself in the black-body formalism and assumes an isotropic emission which directly depends on the local quasi-Fermi level splitting. Since the emissivity is linked to the absorptivity, it also allows us to derive the reabsorption of the emitted flux (PR), notably accounting for possible total internal reflexions. A similar model was employed to assess the trapping of PL photons inside $CsPbBr_3$ micro-wires[46]. We complete their analysis by separating the monitored $I_{PL}$ signal into a propagated and a locally emitted component, which allows us to characterize short-range and long-range transport properties. Another technique to account for this photon flux consists in transforming the continuity equation for electrons and holes into a coupled differential system with various photon fluxes[8,32]. This was mainly used to predict long-range photon recycling but not to separate short-range electronic and photonic transport. In the present study, we do so by using a single differential equation summarizing the photon diffusion in a unique regeneration term, which results in a simple model adapted to fitting multiple parameters at once.

We have shown that both charge carrier diffusion (led by mobility and lifetime) and photon diffusion (led by absorption properties) contribute to the transport. It is important to know the weight of PR in the transport as it has a beneficial effect on $V_{oc}$[5]. Nevertheless, it gets mitigated if light management happens, as it leads to a better light in- and out-coupling. Indeed, the escape probability of PL photons is directly included in the formula giving the $V_{oc}$ boost induced by PR[47]. Various models for photon propagation inside a semiconductor thin film coexist. They assume either an angular randomization at each reflection, which leads to the famous "4n$^2$ limit" derived by Yablonovich[48], or a reflection as of Snell-Descartes law. The calculation we propose is based on the latter, in which 80% of the photons propagates in the 600 nm-thick film via guided modes[49]. It reproduces very well the deformation of PL spectra at long range. However, a small amount of outcoupling of these guided modes is necessary for us to observe the photon propagation flux.

The obtained electronic and photonic parameters are in line with the values one can find separately in the literature. In our work, the monomolecular lifetimes $\tau_n$ values are effective lifetimes and take into account the non-radiative recombination phenomena occurring at the bulk and at the interfaces. This explains their low values, in comparison to studies where they refer only to the bulk[11,25]. For what concerns $R_{eh}^\star$, its value is not only dependent on the material quality, but also on the geometry of the excitation profile and of the thin film itself. Once corrected with $g_{corr} = 0.43$, it leads to the internal radiative coefficient $R_{eh}$, which closely matches calculated ones[32], and measured ones[31,50]. Being an intrinsic material property, it also allows to determine the charge carrier intrinsic concentration $n_i^2$ ($4.4 \times 10^{14}$ cm$^{-3}$ according to Supplementary Equation 3).

As previously mentioned in the introduction a large range of transport parameters might be found for perovskite absorbers. Besides the different sample qualities and compositions, one origin might be found in the concomitant electronic and photonic contribution investigated in this paper. On the one hand, previous works have underlined the photon propagation as well as the PR and we here confirm these results[8,26,27,46]. However, so far no complete model was developed in order to determine the transport properties. On the other hand, several experiments based on luminescence analysis (especially time-dependent experiments) measure the carrier lifetime and transport properties. Nevertheless, they do not take into account the two photonic contributions[10,12,16,17,27,51]. This fact goes along with pump-probe experiments where the spatial contribution probes the solely carrier diffusion[22,43,52]. Although the photon propagation can be ignored in the latter experiment, the PR affects the carrier concentration and should be considered. Those experiments derived fairly high values for $D_n$[43,52], which are an order of magnitude higher than observed in this study. In a general way, luminescence experiments based on point pulsed illumination in perovskite might deliver lifetime values not so much impacted by PR[51], but the diffusion away from the excitation spot remains influenced by PR and it is not straightforward to characterize carrier transport with it. A recent study illustrates this artifact where the authors determined $D_n$ by fitting the time-resolved PL signal away from the excitation and attributed the whole dynamics to the electronic diffusion, which could yield an overestimated determination of $D_n > 1$ cm$^2$ s$^{-1}$[53]. Also, we want to highlight here a simple idea to monitor solely the direct emission and not the propagated one. This consists in adding a color filter at $E_{peak}$ in the collection branch to get rid of long-range photons, which allows then to distinguish electronic and photonic diffusion,

In conclusion, transport in a semiconductor thin film happens via two channels. The first one consists in electron and hole diffusion through scattering and it can be described by a diffusion coefficient, which takes low values in metal halide perovskite. Henceforth, this transport is essentially short-range. The photon recycling constitutes the second one and can be observed at any range. Indeed, it does not only induce long-range transport, but also significantly influences short-range transport. Thanks to time-resolved PL profiles acquired after a pulsed point illumination, we have been able to decorrelate both electronic and photonic regimes and to quantify their impact on charge carrier transport. Additionally, a photon propagation regime was observed, confirming the guided propagation of PL photons inside the thin film. In a nutshell, we determine precise values for bulk-related parameters, namely the charge carrier mobility and monomolecular lifetime, the external radiative coefficient, as well as for device-related light management parameters, namely the photon escape probability and the long-range photon recycling length. The introduced contactless experiment can find interest in any semiconducting materials where both electronic and photonic contribution happen, especially on high-quality materials.

## Methods

**Sample fabrication.** Perovskite film deposition. The deposition solution was prepared by dissolving 1.10 M PbI$_2$ (TCI Chemicals), 0.20 M PbBr$_2$ (Alfa Aesar), 1.00 M formamidinium iodide (FAI, Dyesol) and 0.20 M methyl ammonium bromide (MABr, Dyesol) in a mixture of DMSO:DMF (4:1 in v/v) as solvent. The Cs inorganic cations is added from a stock precursor solutions of CsI (Sigma Aldrich) 1.50 M in DMSO. Perovskite solutions were deposited by spin coating using a double plateau (2000 rpm to cast the precursor solution followed by 6000 rpm to drip 100 μL of chlorobenzene). Then, the samples were annealed at 100 °C during 30 min. The previously described films were deposited on a bare glass cleaned with UV-Ozone to enhance its wettability. A photovoltaic device has been built using this 600-nm thick material as absorber in the following stack: FTO/TiO$_2$/Perovskite/Spiro-Ometad. The intrinsic quality of the bulk was evidenced by stabilized efficiencies of 18.8%, with an open-circuit voltage of 1.1 V[35].

**Experimental setups.** The hyperspectral imaging system records luminescence intensity signal along three dimensions $\{x,y,\lambda\}$. Its mains components are: a home built microscope with Thorlabs optomechanical elements, a 2D bandpass filtering system from company PhotonEtc with 2 nm resolution and finally a 1Mpix silicon-based CCD camera PCO1300. The sample is illuminated through an infinity-corrected X100 Nikon objective with numerical aperture 0.95 and the luminescence is collected through the same objective. The excitation beam and luminescence signals are separated with appropriate dichroic beamsplitter and filters. The 2D luminescence signal is corrected for each pixel of the sensor from the spectral transmissions along all the optical path, from the read noise and dark current noise

of the camera, and from the spatial inhomogeneity of the optical system (flat-field). The spatial resolution is 1 μm.

The Time-Resolved-FLuorescence IMaging setup records luminescence intensity signal along three dimensions {x,y,time}. The sample is illuminated with a homemade microscope (Thorlabs components) through the same Nikon objective X100 with $NA = 0.95$ and the luminescence signal collected also through the same objective with appropriate dichroic beamsplitter and filter to eliminate the 550 nm band. The images are acquired with a gated intensified Em-iCCD camera Pimax-4 from Princeton Instrument, triggered by the laser itself. The temporal gates are being progressively delayed from the beginning of the laser pulse until the end of the luminescence decay. The minimum gate width of the camera is 480 ps which allows a temporal resolution of the system about 750 ps. The spatial resolution is 1 μm.

For both setup, the optical excitation is made with a supercontinuum Fianium Laser set at 550 nm with pulse width 6 ps, and repetition rate of 1 MHz. The diffraction limit in these conditions is 350 nm. We have evaluated by measurement of the OTF (Optical Transfert Function) that the spot has a diameter of 950 ± 50 nm. The absorption coefficient of perovskite was determined using a combination of transmission/reflexion measurements at photon energies above the bandgap and fourier transform photocurrent spectroscopy at all energies, which are absolutely calibrated thank the first ones. The resulting absorption coefficient measurement is displayed in Supplementary Fig. 10.

**Modeling**. Modeling of lateral diffusion and recombination of charge carrier concentration is realized using Matlab, and Curve Fitting toolbox. The differential equation is described using pdepe function. The solver provides optimal solutions (in log scale) and confidence intervals for each parameter of the fit. The initial condition is a Gaussian charge carrier distribution, which FWHM is a fitting parameter.

## Data availability

The authors declare that all data supporting the findings of this study are available within the paper and the Supplementary Information, or available from the authors upon request to A.B. (Adrien.bercegol@edf.fr) or L.L. (Laurent.lombez@cnrs.fr)

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

## Acknowledgements

This project has been supported by the French Government in the frame of the program of investment for the future (Programme d'Investissement d'Avenir - ANR-IEED-002-01). Amelle Rebai, Armelle Yaiche, and Aurélien Duchatelet are acknowledged for perovskite films deposition and PV device fabrication. Christophe Longeaud is acknowledged for absorption coefficient measurement.

## Author contribution

L.L., D.O., and A.B. conceived and planned the experiments. J.R. fabricated the sample and performed basic characterization. A.B. realized the luminescence experiments, with help from L.L., S.C., D.O., and O.F. A.B., L.L., D.O., and D.S developed the radiometric model assessing from photon propagation. A.B. post-treated the acquisitions to fit physical parameters, along with their confidence intervals. Data interpretation was realized by A.B., L.L., and D.O. A.B., L.L., and S.C. wrote the manuscript in close consultation with other authors.

## Additional information

**Competing interests:** The authors declare no competing interests.

