## [Peer Review File · Nature Communications]

Reviewers' comments:

Reviewer #1 (Remarks to the Author):

This work presents an experimental technique in order to measure the transport properties of the highly efficient light absorbing layers. Among these layers are halide perovskite thin films that have attracted widespread interest in recent years. It has been revealed that reabsorption of the emitted photons inside these layers (the so-called photon recycling effect) can affect the transport and recombination kinetics of the photogenerated carriers. Therefore, a correct decoupling of the photon and charge transport properties is necessary to achieve a reliable assessment of the photonic and electronic properties of the layer. This manuscript, in fact, addresses this important issue. In order to quantify the quantities mentioned above, two experimental techniques are used together: a steady state hyperspectral imaging system and a time-resolved fluorescence imaging system. As far as I know, this combined experimental setup has not been utilized in previous works addressing the photon recycling effect. The discussions and experimental evidence given in this manuscript are interesting and timely for the community. Therefore, I can recommend this manuscript for publication. However, several mandatory revisions are required and the following points should be addressed clearly by the authors.

1. Regarding the theoretical modeling the photon recycling effect, the authors claim that "*However, a model able to unambiguously decorrelate electronic and photonic properties is still missing.*". As a matter of fact, it should be noted that three techniques have been used in the literature in order to model the photon reabsorption effect:

- a) ray optical modeling; JAP, 2013, 113, 123109 & NanoLett, 2017, 17, 5782
- b) photon diffusion model; Science, 2016, 351,1430 & PRAppl, 2018, 10, 034062
- c) detailed balance with black body radiation; JPCL, 2017, 8, 5084 & PRL, 2018, 120, 057404 & ACS Energy Lett., 2018, 3, 1492

Here, the authors use the third approach. They introduce a generation term in the continuity equation that accounts for the reabsorption of the emitted photons. this method, which is based on the black body formalism, has already been used by Staub et. al. (JPCL, 2017, 8, 5084) and Yamada et. al. (PRL, 2018, 120, 057404) in very recent works. Therefore, it should be clarified whether and in what respect their modeling is different from those works, and what is missing here.

2. No clear physical explanation is provided by the authors for the value of the transition distance which determines the transition between two regimes in Figure 2 ($r \approx 3 \mu\text{m}$). It seems that this value is originated from the absorption coefficient of the layer shown in Figure S8. Therefore, it is recommended that instead of $r \geq 3 \mu\text{m}$ in Table 1, the authors use $r \geq \alpha_{\text{peak}}^{-1}$ to separate the different regimes. If there is an alternative explanation, it should be provided for the readers.

3. As we know, electronic diffusion length is determined by the total lifetime, that is, $1/(\tau_n^{-1} + R_{eh} n)$, and not merely by SRH lifetime τ_n . Consequently, the equivalent

electronic diffusion should be calculated using the relation $\sqrt{D_n/(\tau_n^{-1} + R_{eh} n)}$. But, the value estimated in the manuscript is about 450 nm that seems to be calculated using $\sqrt{D_n \tau_n}$. Do the authors confirm that the local density of electrons is low enough to ensure $(\tau_n^{-1} + R_{eh} n)^{-1} \approx \tau_n$? In addition, it is important to assure readers that the density is also low enough to neglect the Auger recombination term in the continuity equation as done in equation 3.

4. For $r > 3\mu\text{m}$, the authors assume that $I_{PL}(E < E_{peak}) = \phi_{prop}$, meaning that it is assumed that $\phi_{prop} \gg R_{eh}^* n^2$ in this regime. However, I do not find any quantitative justification for this assumption in the manuscript. It is seen in Figure 2c that the PL intensity does not substantially differ for $E < E_{peak}$ and $E > E_{peak}$. Therefore, if $I_{PL}(E > E_{peak}) = R_{eh}^* n^2$, why we should not expect a similar behaviour for $I_{PL}(E < E_{peak})$? In other words, according to the approximately symmetric shape of the luminescence spectrum, $R_{eh}^* n^2$ is expected to be responsible also for $E < E_{peak}$, or at least $R_{eh}^* n^2 \sim \phi_{prop}$.

5. Regarding the photonic diffusion length, it is said in the manuscript that "... an effective photonic diffusion length L_{PR} can be extracted from PL profiles at $E > E_{peak}$ where the PL profile decays with characteristic length of $2.5\mu\text{m}$. Knowing that the PL decreases quadratically with the carrier concentration n , L_{PR} is measured around $5\mu\text{m}$." As photons of different energies have different absorption lengths, can a single photonic diffusion length be assigned to the electrons? I think the authors should comment on this point briefly.

6. There seems to be a confusion between the polar and the spherical coordinates in the manuscript. Since the authors use the polar coordinates in their modeling, the intensity attenuation should be considered as $1/r$ and not $1/r^2$ as done throughout the modeling.

7. After initial excitation, photogenerated carriers spread in the layer and consequently FWHM of the carrier distribution increases. On the other hand, as can be seen in Figure S2, the correction factor g_{corr} decreases as FWHM increases. Therefore one can conclude that $g_{corr} = g_{corr}(t)$. But this time dependency is not considered in the modeling and a constant $g_{corr} = 0.35$ is used by the authors. Does correction factor stay constant around this value as time goes on, whereas FWHM increases with time?

8. Quality of some figures in the main text and supplementary Information is very poor, for example, Figure 2s which is an important figure. Some parts of the manuscript are also difficult to understand because the necessary explanations are not given. For example, an explicit formula for $g_{rec}(r)$ is not provided by the authors. I

also strongly recommend that the authors use r' instead of r for all quantities that are going to be integrated over the distance.

9. Typo: line 220 $10^{10} \rightarrow 10^{-10}$

Reviewer #2 (Remarks to the Author):

The authors propose an optical method to measure different transport properties of a semiconductors. The method is applied to halide perovskites. They have obtained the carrier diffusion properties, taking into account the effect of photon recycling (PR) and photon propagation. Optical experiments have been proposed previously to extract the transport properties of halide perovskite, but none has taken into account PR and photon propagation, which lead notably to an overestimation of carrier diffusion.

The agreement between the experimental data and the model is very good and convincing. The manuscript is clear and well-written. The subject of the manuscript will very likely appeal to the broad audience of Nature Communications. For these reasons, I recommend the manuscript for publication in Nature Communications after some minor corrections.

Comments:

1. I expect the size of excitation spot to be sub-micrometric, ie small in comparison with the typical distance studied here. However, I think the authors should mention this size in the manuscript.
2. It would be better if figures appear in the order they are commented in the text. Figure 2b and 2c should be inverted.
3. There is a problem with the inset legend in figure 2c.
4. Line 209 and 212, there are mistakes in the numbering of figures.
5. Line 220 Reh should be to the power -10 not +10.
6. Figures in Supplementary Information (SI) are not readable (Figure S2)

Reviewer #3 (Remarks to the Author):

This manuscript by Bercegol et al. reported an optical method to differentiate the electronic contribution from the photonic contribution to the carrier diffusion length in halide perovskite thin film.

Particularly, they discovered that the photon propagation, which has rarely been considered in previous reports, plays a significant role in photon transport in halide perovskites in addition to the photon recycling. Therefore, they derived the accurate carrier diffusion coefficient, monomolecular lifetime, and radiative coefficient based on modeling with the consideration of the photonic contribution to the carrier transport.

Overall, I think this is a high quality paper that deserves a publication in Nature Communications, and the results are rather important for the understanding of the carrier transport properties in lead halide perovskites.

Based on above consideration, I suggest the paper can be accepted after minor revision and the authors are suggested to do the excitation intensity dependent PL measurements to further verify the accuracy of the proposed model.

Referee 1

This work presents an experimental technique in order to measure the transport properties of the highly efficient light absorbing layers. Among these layers are halide perovskite thin films that have attracted widespread interest in recent years. It has been revealed that reabsorption of the emitted photons inside these layers (the so-called photon recycling effect) can affect the transport and recombination kinetics of the photogenerated carriers. Therefore, a correct decoupling of the photon and charge transport properties is necessary to achieve a reliable assessment of the photonic and electronic properties of the layer. This manuscript, in fact, addresses this important issue. In order to quantify the quantities mentioned above, two experimental techniques are used together: a steady state hyperspectral imaging system and a time-resolved fluorescence imaging system. As far as I know, this combined experimental setup has not been utilized in previous works addressing the photon recycling effect. The discussions and experimental evidence given in this manuscript are interesting and timely for the community. Therefore, I can recommend this manuscript for publication. However, several mandatory revisions are required and the following points should be addressed clearly by the authors.

1. Regarding the theoretical modeling the photon recycling effect, the authors claim that "*However, a model able to unambiguously decorrelate electronic and photonic properties is still missing.*". As a matter of fact, it should be noted that three techniques have been used in the literature in order to model the photon reabsorption effect: a) ray optical modeling; JAP, 2013, 113, 123109 & NanoLett, 2017, 17, 5782 b) photon diffusion model; Science, 2016, 351,1430 & PRAppl, 2018, 10, 034062 c) detailed balance with black body radiation; JPCL, 2017, 8, 5084 & PRL, 2018, 120, 057404 & ACS Energy Lett., 2018, 3, 1492. Here, the authors use the third approach. They introduce a generation term in the continuity equation that accounts for the reabsorption of the emitted photons. This method, which is based on the black body formalism, has already been used by Staub et. al. (JPCL, 2017, 8, 5084) and Yamada et al. (PRL, 2018, 120, 057404) in very recent works. Therefore, it should be clarified whether and in what respect their modeling is different from those works, and what is missing here.

We thank the referee for providing his insight on literature dealing with PL reabsorption. Some references were already cited in the paper¹⁻⁵, while the other ones have now been added^{6,7}. In the following, we explicitly describe the main advances allowed by our model, which is valid for pulsed inhomogeneous illumination. Thanks to this fact, we decorrelate for the first time electronic and photonic contributions to transport at any range. The comparison with the others existing models has been added in the discussion part of the article.

- Ray optics models have been used in 1-dimensional for steady-state illumination⁶, or pulsed illumination¹. They allow quantifying the recycling ratio at a global scale, without taking into account inhomogeneities (from the illumination or from the material). They are now mentioned in the introduction, as well as in the discussion. They have not been employed to separate electronic and photonic transport.
- Photon diffusion model have been introduced by Pazos-Outon et al.⁸ and its numerous applications have been studied numerically by Ansari-Rad et al.³. As it was written in the discussion "They consist in transforming the continuity equation for electrons and holes into a coupled differential system with various photon fluxes^{3,8}." We have now added that they were mainly used to predict long-range photon recycling but not to separate short-range electronic and photonic transport.
- Then, models including emission proportional to the generalized Planck law have been considered^{4,5}. They are in principle similar to our study, but only apply to a homogeneous

illumination. Staub et al. consider TRPL and spectral data to deduce a recombination model for charge carrier⁴. In the work of Yamada et al., the excitation depth is enlarged to induce a red-shift of the spectrum⁵. Both do not apply for inhomogeneous illumination and hence do not bring information regarding charge carrier transport. They are mentioned in the introduction and in the discussion as well.

- The case of the paper by Dursun et al. is slightly different, it includes the spatial component and explains the progressive red-shift of I_{PL} away from the excitation spot in a CsPbBr₃ micro-wire. Their kinetics model for charge carrier takes into account every phenomenon we consider, except Beer-Lambert attenuation. The paper is mainly focus on the long distance contribution where the authors believe in the existence of an efficient photon recycling coupled to a high luminescence quantum efficiency (requirement for long range contribution). There is no fitting procedure in order to determine transport properties as it is not the main objective of the papers. Also, the authors apply their kinetic model at long distance from the excitation spot where it might contain Φ_{prop} in addition to a direct contribution proportional to the local charge carrier concentration. These considerations have been added to the discussion section of the article.

2. No clear physical explanation is provided by the authors for the value of the transition distance which determines the transition between two regimes in Figure 2 ($r \approx 3 \mu\text{m}$). It seems that this value is originated from the absorption coefficient of the layer shown in Figure S8. Therefore, it is recommended that instead of $r \geq 3 \mu\text{m}$ in Table 1, the authors use $r \geq \alpha_{peak}^{-1}$ to separate the different regimes. If there is an alternative explanation, it should be provided for the readers.

Figure R1 (reproducing figure 2 of the article) (a) spectrally-resolved and (b) time-resolved PL profiles following a point pulsed illumination. This figure evidences the delimitation point between PL regime and photon propagation regime.

Indeed, there is a link between the transition value and α_{peak}^{-1} as the the order of magnitude matches : $\alpha_{1.57eV} = (3\mu\text{m})^{-1}$. A comment has been added to the text. However, this limit is related to the intrinsic electronic property and linked to geometry of the problem (thickness of sample, width of the excitation profile). Therefore we still mention that the value was defined empirically from the shape of the spectrally-resolved PL profiles by looking at Figure R1-A (similar to Figure 2-A). Note that the answer to the point 4) (see below) brings further insight into the progressive appearance of propagated signal into the total one.

3. As we know, electronic diffusion length is determined by the total lifetime, that is, $1/(\tau_n^{-1} + R_{eh} n)$, and not merely by SRH lifetime τ_n . Consequently, the equivalent electronic diffusion should be

calculated using the relation $\sqrt{D_n/(\tau_n^{-1} + R_{eh} n)}$. But, the value estimated in the manuscript is about 450 nm that seems to be calculated using $\sqrt{D_n \tau_n}$. Do the authors confirm that the local density of electrons is low enough to ensure $(\tau_n^{-1} + R_{eh} n)^{-1} \approx \tau_n$? In addition, it is important to assure readers that the density is also low enough to neglect the Auger recombination term in the continuity equation as done in equation 3.

Regarding the calculation of L_n , it has been done for $10^{16}/\text{cm}^3$. This concentration value is representative for 1 sun operation of perovskite photovoltaic. This calculation took into account the radiative lifetime $\frac{1}{n R_{eh}} = 660 \text{ ns}$ as well as the non-radiative one $\tau_n = 110 \text{ ns}$. Such a precision was added to the main text.

Auger recombination takes a significant part in the continuity equation once the charge carrier concentration is above $10^{18} /\text{cm}^3$ according to various sources^{3,9}. In our study, fits realized at low injection are not impacted. For fits realized at high injection, the concentration of charge carriers after 10 ns, when the fitting procedure begins, evaluates at $4 \times 10^{17}/\text{cm}^3$, while it elevates at $10^{18} /\text{cm}^3$ at $t=0\text{ns}$, and drops to $10^{17} /\text{cm}^3$ at $t=50\text{ns}$. These numbers are given according to simulations realized with the parameters fitted in our study. To ensure that Auger recombination does not perturbate our kinetics analysis significantly, fits at high injection were realized again with coefficients in the order of magnitude described by the literature (from 10^{-29} to $5 \cdot 10^{-28} \text{ cm}^6/\text{s}$). This is displayed in Figure R2, where the title of each subplot indicates the value determined for D_n , R_{eh}^* and τ_n . We conclude that Auger recombination have a very small impact on the determined coefficients, especially the SRH lifetime and the diffusion coefficient which remain constant. If even higher injection were used, a more complete version of the continuity equation should be used to fit the results. A dedicated section in the Supplementary Information was added.

Figure R2 Fitting the $I_{PL}(r,t)$ surface obtained at high fluence with the space and time-domain restriction, and accounting for Auger recombination. Various Auger coefficients C_{auger} are taken [a: 10^{-29} ; b: 5×10^{-29} ; c: 9×10^{-29} ; d: 1.3×10^{-28} cm^6/s] in the range of literature values^{4,10}. The impact on lifetime and diffusion coefficients values determined is negligible. The external radiative coefficient determined varies from 4.5 (a) to 2.5 (d) $\times 10^{-11}$ cm^3/s .

4. For $r > 3 \mu\text{m}$, the authors assume that $I_{PL}(E < E_{peak}) = \Phi_{prop}$, meaning that it is assumed that $\Phi_{prop} \gg R_{eh}^* n^2$ in this regime. However, I do not find any quantitative justification for this assumption in the manuscript. It is seen in Figure 2c that the PL intensity does not substantially differ for $E < E_{peak}$ and $E > E_{peak}$. Therefore, if $I_{PL}(E > E_{peak}) = R_{eh}^* n^2$, why we should not expect a similar behaviour for $I_{PL}(E < E_{peak})$? In other words, according to the approximately symmetric shape of the luminescence spectrum, $R_{eh}^* n^2$ is expected to be responsible also for $E < E_{peak}$, or at least $R_{eh}^* n^2 \sim \Phi_{prop}$.

The referee is raising here an important point on the spectral shape influence. We do agree with the comment and clarify that point to facilitate the understanding.

We investigated the impact of the propagated Φ_{prop} on the shape of PL spectra at each $r > 3 \mu\text{m}$, value from which the long-range calculation of PL propagation is valid (formulas (ES11-ES15)). We fitted the spectra as the sum of a spectrum proportional to the signal at $r=0$, representative for the local emission, and of a propagated spectrum at distance r , which was calculated using formula (ES15) :

$$\frac{I_{PL}(r)}{\max_E(I_{PL}(r))} = \beta \frac{\Phi_{prop}(r)}{\max_E \Phi_{prop}(r)} + (1 - \beta) \frac{I_{PL}(r=0)}{\max_E I_{PL}(r=0)}$$

Normalizations are employed so that the sum of coefficients for both contributions is 1, allowing to estimate their relative importance. Results are displayed in Figure R3. The relative weight of direct and waveguided contribution is indicated in the legend of each subplot. The weight of direct emission is 100% at 3 μm and decreases to 40% at $r=9\mu\text{m}$. The weight of propagated spectrum follows an opposite trend.

Figure R3 Reconstruction of I_{PL} signal for increasing distances to the excitation point. The relative weight of the propagated contribution increases from 10 % at $r=4\mu\text{m}$ to 54% at $r=9\mu\text{m}$.

This new outlook of our datasets contains a richer information that the previously observed spectral shift. It clarifies the point raised by the referee such that $R_{eh}^* n^2 \approx \Phi_{prop}$ for $E < E_{peak}$ and we modified the table 1 accordingly. The conclusion we draw out of this analysis remains unchanged : we can unambiguously relate the $I_{PL}(r < 3\mu\text{m})$ to the local charge carrier concentration, while further spectral analysis is required to investigate transport properties using $I_{PL}(r > 3\mu\text{m})$.

r : Distance to excitation spot	Spectrally-resolved $I_{PL}(E < E_{peak})$	Spectrally-resolved $I_{PL}(E > E_{peak})$	Time-resolved $I_{PL}(t)$
$r < 3\mu\text{m}$	$I_{PL} = R_{eh}^* n^2$	$I_{PL} = R_{eh}^* n^2$	$I_{PL} = R_{eh}^* n^2$
$r > 3\mu\text{m}$	$I_{PL} = \phi_{prop} + R_{eh}^* n^2$	$I_{PL} = R_{eh}^* n^2$	$I_{PL} = \phi_{prop} + R_{eh}^* n^2$

The Table 1 was modified accordingly, as indicated in Table R1, and Figure R3 has been added to the Supplementary Information. The text related to this table in the manuscript (section “evidencing the electronic and photonic regime”) was also modified to take into account these new inputs. A reconstruction of PL spectra was incorporated as the inset in Fig2C, while the whole figure R3 is displayed in the Si. (Figure S5)

5. Regarding the photonic diffusion length, it is said in the manuscript that "... an effective photonic diffusion length L_{PR} can be extracted from PL profiles at $E > E_{peak}$ where the PL profile decays with characteristic length of 2.5 μm . Knowing that the PL decreases quadratically with the carrier concentration n , L_{PR} is measured around 5 μm ." As photons of different energies have different absorption lengths, can a single photonic diffusion length be assigned to the electrons? I think the authors should comment on this point briefly.

We thank the referee for this constructive remark. We agree that several photonic diffusion lengths could be defined with a little physical meaning. We wanted to qualitatively describe the progressive attenuation of Φ_{prop} as the one of $I_{PL}(E < E_{peak})$, while each photon energy has its own attenuation length described by its absorption coefficient. Due to the considerations presented in the answer to point 4., $I_{PL}(E < E_{peak}) \neq \Phi_{prop}$ and this analysis is not valid any more. On the contrary we keep it for the photon recycling, as $I_{PL}(E > E_{peak})$ can unambiguously be ascribed to local emission.

6. There seems to be a confusion between the polar and the spherical coordinates in the manuscript. Since the authors use the polar coordinates in their modeling, the intensity attenuation should be considered as $1/r$ and not $1/r^2$ as done throughout the modeling.

We would like to make this point more clear. The $1/r^2$ does not account for the polar/spherical coordinate. It comes from the calculation of the solid angle $d\Omega_{em} = dS_o/r^2$. However, the elementary volume $dS_{em} = r'd\theta'dz$ also contains r' , and hence the final calculation in (ES7) is attenuated with a $1/r'$ factor.

7. After initial excitation, photogenerated carriers spread in the layer and consequently FWHM of the carrier distribution increases. On the other hand, as can be seen in Figure S2, the correction factor g_{corr} decreases as FWHM increases. Therefore one can conclude that $g_{corr} = g_{corr}(t)$. But this time dependency is not considered in the modeling and a constant $g_{corr} = 0.35$ is used by the authors. Does correction factor stay constant around this value as time goes on, whereas FWHM increases with time?

We thank the referee for underlining this very crucial point. We enhanced the quality of Figure S2, and would like to comment it again here under.

First, g_{corr} is mainly derived with geometric and absorption parameters only, so that the injection level has no effect on its value. To study its time dependence, we calculate it for various spatial distribution of charge carriers. Indeed, as time goes on, the spatial distribution widens. Hence, we derive the $g_{corr}(r)$ profiles for various gaussian distributions with increasing FWHM. In Figure R4-a, the local emission and recycling are represented for various cases. In Figure R4-b, g_{corr} profiles are displayed and one can see that the mean value for g_{corr} over $r=[0,3]\mu\text{m}$ is 0.43, with a standard deviation of 0.05 at most when FWHM takes its values between 1 and 3 μm (representative for the evolution of the charge carrier profile during our experiment). Here, it shall be noted that the mean value of g_{corr} evolved from 0.35 ± 0.05 (previous version) to 0.43 ± 0.05 (revised version), as the calculation was done again with a more precise mesh for energetic and spatial scale. It induced changes on the measured R_{eh} values, as the proportionality factor linking R_{eh} to R_{eh}^* was g_{corr} .

The conclusion of this graph is the following. We proved that recycling is proportional to local emission with a constant proportionality factor in our time- and space-restricted dataset. Then, it is not necessary to take into account variations of g_{corr} with time.

Figure R4 (a) Contributions to rate equation (E3) for a gaussian charge carrier distribution with FWHM=1.3μm, and $\Phi_0 = 5 \times 10^6$ ph/pulse. Polychromatic PL is considered (no α_{pR}). The PL emission is displayed in dotted lines, while the recycling is in plain lines. Vertical scale is in arbitrary units. (b) The local R_{eh} correction factor, as defined in (E4) is displayed for gaussian charge carrier distributions with FWHM=1..3μm.

8. Quality of some figures in the main text and supplementary Information is very poor, for example, Figure 2s which is an important figure. Some parts of the manuscript are also difficult to understand because the necessary explanations are not given. For example, an explicit formula for $g_{rec}(r)$ is not provided by the authors. I also strongly recommend that the authors use r' instead of r for all quantities that are going to be integrated over the distance.

SI was rewritten and Figure S2 was split in S2 (figure R4 here) and S3 (figure R5 here), each with an enhanced graphic quality. The generic formula of $g_{rec}(r)$ has been added in the main text. All calculations in the SI are using r' from a secondary frame, while r is only employed for the primary frame centered according to the excitation. We thank the referee for this suggestion of presentation.

$$g_{rec}(r) = \iiint_{(r',\theta',z') \text{ frame centered in } (r,0,0)} \frac{K_p n^2(r',\theta')}{n_0 p_0} \int_E \alpha^2(E) E^2 \exp\left(-\frac{E}{k_B T}\right) \frac{\exp(-\alpha(E)r')}{r'} dE dr' d\theta' dz'$$

Figure R5 Contribution of each PL wavelength to the recycling. The blue curve gives the sum of all contributions, and matches the red curve in Fig S2-A. The black curve was derived under the assumption of a fixed absorption coefficient for the whole PL emission, as defined in equation (E57).

9. Typo: line 220 $10^{10} \rightarrow 10^{-10}$

OK

Referee 2

The authors propose an optical method to measure different transport properties of a semiconductors. The method is applied to halide perovskites. They have obtained the carrier diffusion properties, taking into account the effect of photon recycling (PR) and photon propagation. Optical experiments have been proposed previously to extract the transport properties of halide perovskite, but none has taken into account PR and photon propagation, which lead notably to an overestimation of carrier diffusion. The agreement between the experimental data and the model is very good and convincing. The manuscript is clear and well-written. The subject of the manuscript will very likely appeal to the broad audience of Nature Communications. For these reasons, I recommend the manuscript for publication in Nature Communications after some minor corrections.

We thank a lot the referee for these very encouraging comments on our work, please find below the answer to the questions and suggestions that have been raised.

Comments:

1. I expect the size of excitation spot to be sub-micrometric, ie small in comparison with the typical distance studied here. However, I think the authors should mention this size in the manuscript.

Spot size is sub-micrometric as we use a NA=0.95 objective with a 550-nm wavelength laser. The diffraction limit in these conditions is 350nm. We have evaluated by measurement of the OTF (Optical Transfer Function) that the spot has a diameter of 950+/-50 nm. As the laser excitation might spread as it is absorbed in the depth of the material, the FWHM of the initial charge carrier concentration is taken as a fitting parameter in the fitting code. It is now mentioned in the manuscript (Experimental section).

2. It would be better if figures appear in the order they are commented in the text. Figure 2b and 2c should be inverted.

We inverted the numbering.

3. There is a problem with the inset legend in figure 2c.

The inset was changed for a figure showing the propagated and direct contribution.

4. Line 209 and 212, there are mistakes in the numbering of figures.

Thank you it is now corrected as well.

5. Line 220 Reh should be to the power -10 not +10.

Thank you, we corrected this value.

6. Figures in Supplementary Information (SI) are not readable (Figure S2)

Figure S2 was split in Figure S2 and S3, with enhanced graphic quality.

Referee 3

This manuscript by Bercegol et al. reported an optical method to differentiate the electronic contribution from the photonic contribution to the carrier diffusion length in halide perovskite thin film.

Particularly, they discovered that the photon propagation, which has rarely been considered in previous reports, plays a significant role in photon transport in halide perovskites in addition to the photon recycling. Therefore, they derived the accurate carrier diffusion coefficient, monomolecular lifetime, and radiative coefficient based on modeling with the consideration of the photonic contribution to the carrier transport.

Overall, I think this is a high quality paper that deserves a publication in Nature Communications, and the results are rather important for the understanding of the carrier transport properties in lead halide perovskites. Based on above consideration, I suggest the paper can be accepted after minor revision and the authors are suggested to do the excitation intensity dependent PL measurements to further verify the accuracy of the proposed model.

Figure R6 (left) PL spectra obtained for increasing fluence with an illuminated surface of approx. $60\mu\text{m}^2$ with a green continuous wave laser, represented in absolute units. (right) Normalized spectra underlining the stability of its shape with power.

We thank the referee for this remark. We realized this power study with a green continuous-wave laser illuminating a triple cation perovskite sample deposited on glass. Hyperspectral images were acquired and a mean spectrum was calculated on the illuminated zone ($\approx 60\mu\text{m}^2$). The spectra displayed in Figure R6-A do not shift as the injection is reduced. This appears even more clearly in Figure R6-B, where they are normalized. Henceforth, power-induced effect cannot explain the spectral shift observed in Figure 2-AB.

This figure was added in the Supplementary Information (section V), and a reference is made to it in the section ‘evidencing the electronic and photonic regime’.

References

1. Crothers, T. W. *et al.* Photon Re-Absorption Masks Intrinsic Bimolecular Charge-Carrier Recombination in $\text{CH}_3\text{NH}_3\text{PbI}_3$ Perovskite. *Nano Lett.* **17**, 2834–2839 (2017). doi:10.1021/acs.nanolett.7b02834
2. Pazos-Outón, L. M. *et al.* Photon recycling in lead iodide perovskite solar cells. *Science* (80-.). **351**, 1430–1433 (2016).
3. Ansari-Rad, M. & Bisquert, J. Insight into Photon Recycling in Perovskite Semiconductors from the Concept of Photon Diffusion. *Phys. Rev. Appl.* **10**, 034062 (2018).
4. Staub, F., Kirchartz, T., Bittkau, K. & Rau, U. Manipulating the Net Radiative Recombination Rate in Lead Halide Perovskite Films by Modification of Light Outcoupling. *J. Phys. Chem. Lett.* **8**, 5084–5090 (2017).
5. Yamada, T., Aharen, T. & Kanemitsu, Y. Near-Band-Edge Optical Responses of $\text{CH}_3\text{NH}_3\text{PbCl}_3$ Single Crystals: Photon Recycling of Excitonic Luminescence. *Phys. Rev. Lett.* **120**, 57404 (2018).
6. Steiner, M. A. *et al.* Optical enhancement of the open-circuit voltage in high quality GaAs solar cells. *J. Appl. Phys.* **113**, 0–11 (2013).

7. Dursun, I. *et al.* Efficient Photon Recycling and Radiation Trapping in Cesium Lead Halide Perovskite Waveguides. *ACS Energy Lett.* **3**, 1492–1498 (2018).
8. Pazos-Outon, L. M. *et al.* Photon recycling in lead iodide perovskite solar cells. *Science* (80-.). **351**, 1430–1434 (2016).
9. Braly, I. L. *et al.* Hybrid perovskite films approaching the radiative limit with over 90 % photoluminescence quantum efficiency Hybrid Perovskite Films Approaching the Radiative Limit with. (2018).
10. Huang, J., Yuan, Y., Shao, Y. & Yan, Y. Understanding the physical properties of hybrid perovskites for photovoltaic applications. *Nat. Rev. Mater.* **2**, (2017).

REVIEWERS' COMMENTS:

Reviewer #1 (Remarks to the Author):

The changes are satisfactory. Accept as is

Reviewer #2 (Remarks to the Author):

The authors have made the required corrections to the manuscript. I recommend the paper for publication in Nature Communications.

Reviewer #3 (Remarks to the Author):

The author has clarified my comments in the revised paper. I do not have more comments and I suggest the paper can be accepted.